# Unhealthy Lifestyles and Retinal Vessel Calibers among Children and Adolescents: A Systematic Review and Meta-Analysis

**DOI:** 10.3390/nu15010150

**Published:** 2022-12-28

**Authors:** Dan-Lin Li, Miao Zhou, Chen-Wei Pan, Dan-Dan Chen, Meng-Jiao Liu

**Affiliations:** 1School of Public Health, Medical of Soochow University, Suzhou 215123, China; 2Department of Ophthalmology, Peking University People’s Hospital, Beijing 100034, China; 3School of Radiation Medicine and Protection, State Key Laboratory of Radiation Medicine and Protection, School for Radiological and Interdisciplinary Sciences (RAD-X), Collaborative Innovation Center of Radiation Medicine of Jiangsu Higher Education Institutions, Soochow University, Suzhou 215123, China; 4Jiangxi Provincial Key Laboratory of Preventive Medicine, School of Public Health, Nanchang University, Nanchang 330006, China

**Keywords:** life style, retinal vessel calibers, meta-analysis, children, adolescents

## Abstract

The retinal vessel caliber (RVC) is an important biomarker of cardiovascular diseases, which can be semi-automatically measured by fundus photography. This review investigated the associations between the RVCs and the life styles of early life, such as physical activity (PA), sedentary behavior (SB), dietary and sleep, by summarizing the findings from studies on children and adolescents. Two databases (Medline and Embase) were searched from their inception to 30 June 2022. The selected studies were literatures on observational designs, fundus photographs, retinal vessels and lifestyles of children or adolescents. Correlation coefficients of unhealthy life styles and RVCs were transformed to Fisher’s z-scores, and the random-effects model was applied to pool data. A total of 18 observational studies were selected; the lifestyles accessed include 9 studies for PA and SB, 8 studies for dietary and 1 study for sleep. The meta-analysis on the correlation coefficients of regression models found the high level of SB (qualified by screen time, ST) was associated the narrower central retinal arteriolar equivalent (CRAE) among children (r = −0.043, 95% confidence intervals [CI] −0.078 to −0.009). By comparing the first and fourth quartiles of PA, the meta-analysis showed that more indoor PA was associated with smaller venular calibers and more outdoor PA was associated with wider CRAE (r = 0.88, 95%CI −3.33 to 0). Unhealthy lifestyles might be harmful on the retinal microcirculation among children and adolescents but their health effect seems not to be as significant as those in adults.

## 1. Introduction

Atherosclerosis of children in the previous studies has been found to be associated with early signs of coronary heart diseases in adulthood [1,2]. However, due to the lack of hard endpoints for cardiovascular diseases (CVDs) in early life, promising candidate biomarkers are needed to quantify CVD risks in children [3]. The noninvasive observation of retinal micro vessels provides a unique opportunity to assess the impacts of CVD risk factors on microcirculation [4,5,6,7]. Recent evidence has indicated that microvascular alternations of the retinal vessel caliber (RVC), such as retinal arteriole narrowing or venule widening may be potential markers of CVD risks independent of traditional risk factors [4,8]. In addition, the RVC has also been found to be associated with obesity and blood pressure in children [6]. Therefore, the assessment of RVCs in children and adolescents may serve as a promising diagnostic tool for the prevention and the intervention of early CVDs.

The unhealthy lifestyles of prolonged sedentary behavior (SB) and fast food consumption are known to increase the risk of CVDs [6,9,10], whereas regular physical activity (PA) is generally recommended to protect cardiovascular health [11,12]. Interestingly, there is increasing evidence showing that lifestyles may also influence retinal microvascular health across the lifespan. PA could protect the microvascular health through not only increasing the nitric oxide (NO) bioavailability but also lowering the systemic oxidative stress levels [13,14,15]. Things are the opposite for SB. The Sydney Childhood Eye Study demonstrated that more outdoor activities could lead to wider artery calibers, and TV watching resulted in narrower artery calibers [6]. In addition, frequent consumption of takeaway food or high-sugar drinks have been reported to affect retinal micro vessels [10,16]. However, there are few meta-analyses integrating the existing evidence on the association between unhealthy lifestyles and RVCs in the early stage of life.

To fill this gap, we performed a systematic review and meta-analysis on the effects of PA, SB, diet and sleep on retinal blood calibers among children and adolescents. We also performed another separate meta-analysis on statistical changes in sizes of arterioles and venules from the first to fourth quartiles of PA and SB.

## 2. Materials and Methods

We performed the systematic review and meta-analysis following the Meta-analysis Of Observational Studies in Epidemiology (MOOSE) guidelines [17].

### 2.1. Data Sources

We searched PubMed and Embase databases for studies on the associations of lifestyles with retinal microvascular parameters among healthy children and adolescents. The MeSH terms, keywords and search limits for the following three topic areas were combined: (1) retinal microvascular parameters (including retinal vascular calibers and microvasculature); (2) lifestyle (including physical activity, sedentary, food intake and sleep); (3) children or adolescents; and (4) observational studies. The search was limited to all studies published in English through to 30 June 2022.

### 2.2. Study Selection

The screening process was conducted independently by two reviewers (DLL and MZ), and an adjudicator (CWP) made the final decision if any conflict happened. After abstract and title screening, each reviewer independently screened full texts using pre-defined inclusion criteria. Studies were included if they were observational designs, had retinal vessel calibers from fundus photographs, and were about the lifestyles of children or adolescents. We did not included studies performed in patients with hypertension, obesity, or any other CVD-related conditions. The optical coherence tomography (OCT) technology related reports were excluded from the current study even if retinal vessels were analyzed. The additional hand-searching for the reference lists of relevant studies was conducted to ensure the completeness of the selection.

### 2.3. Data Extraction and Quality Assessment

Data were extracted from the included publications by each reviewer independently. These data include first author, year of publication, study population, study name, sample size, participant age, measurement method, statistical method, correlation coefficient, confidence intervals (CI), and estimates of relevant covariates. The models with the minimal adjustments (e.g., age and/or gender adjustments) were extracted to avoid over-adjustment of potential intermediate variables and to improve comparability between studies.

The study quality was assessed from 0 to 10 stars using a modified version of the Ottawa–Newcastle scale checklist [18]. The checklist considers three biases including selection of study population (five stars), comparability between study groups (two stars), and adequacy of results evaluation (three stars). The study quality is defined as high (9–10 stars), medium (7–8 stars), and low (≤6 stars). The evaluation was performed independently by two reviewers (DLL and MZ).

### 2.4. Data Extraction and Quality Assessment

Compared with adult studies on lifestyles and retinal vessels, the studies on children and adolescents are limited. The retinal vessel calibers are usually qualified by central retinal arteriolar equivalent (CRAE), central retinal venule equivalent (CRVE) and arteriole to venule ratio (AVR). To conduct a meta-analysis, we combined correlation coefficients (β) from regression models of CRAE and CRVE among the selected studies. In order to maintain the consistency of regression models, we extracted the results from the models with minimal adjustments, basically only age and/or gender adjustments. The correlation coefficients β were standardized (Sβ), so that the direct comparisons between associations across studies could be conducted and the issue of directionality could be avoided. In this way, the RVC was treated as the outcome measure rather than exposure.

Correlation coefficients were transformed to Fisher’s z-scores, and the random-effects model was applied. The 95%CI of the Fisher’s z-scores were calculated, which were back-transformed to the metric of the correlation coefficients. The I square index (I2) indicates statistical heterogeneity, which was divided into three groups: low (0–25%), moderate (26–74%) or high (≥75%). The R software (version 4.0.5) was used for the meta-analysis.

Another separate meta-analysis was performed on the changes in sizes of arterioles and venules from the first to fourth quartiles of PA or SB. The magnitude and uncertainty of heterogeneity measurements were measured by statistical indices from the Cochran’s Q test. The degree of inconsistency (I2) indicates the magnitude of statistical heterogeneity. The *p* values of the Q statistics were presented for the uncertainty of apparent heterogeneity. The I2 was expressed as a ratio (not directly affected by the study amount), and was usually qualified as small (25%), moderate (50%) and large (75%) totals of inconsistency.

## 3. Results

The detailing numbers of studies searched, screened and included in the review was shown in Figure 1. The systematic review included 18 publications among which 9 studies were about PA or SB [3,6,8,11,12,13,19,20,21], 8 were about diet [7,9,10,16,22,23,24,25] and 1 was about sleep [26]. The characteristics of those studies were shown in Table 1. The studies on PA and SB were cross-sectional and conducted in Europe (3 from Germany [13,20,21], 1 from Denmark [12] and 4 from Switzerland [3,8,11,19]), Singapore [25] or Australia [7,9,10,16,22,23,24,26]. According to the modified version of the Ottawa–Newcastle scale checklist, the study quality for most of the PA and SB studies are medium. The analysis on dietary studies were challenging given the food variety, which was conducted mostly in Australia except one in Singapore. Their study qualities ranged from low to medium.

We conducted the meta-analysis with PA and SB as shown in Figure 2 and Figure 3, where the PA and SB were usually quantified by the hours of the activity or inactivity. The linear regression models of the studies on children’s RVC were classified into the lifestyle factors of PA, SB and diet in Table 2. For the consistency of regression models, the results were extracted from the models with minimal adjustments. The correlation coefficients β were standardized for the direct comparison in the meta-analysis. Among studies of PA models, vigorous PA were considered in particular [8,11,12], apart from the indoor and outdoor activities. Among the SB models, the studies investigated screen time (ST) or television time (TV) separately [3,6,8,11].

Among the dietary models, some studies were focused on the specific ingredients, such as the long chain omega-3 polyunsaturated fatty acid (LCn-3PUFA) [22]. Some addressed the particular types of food, such as high carbohydrates, soft drinks, sugar-sweetened beverages, takeaway food, inflammatory diets and dairy products [7,9,16,24]. Others classified the diet quality or the healthiness of food, and considered the cumulative effects of diet trajectory over a long time [10,23]. The sleep study was about the effect of sleep duration in early childhood on the artery structures of the teenager [26].

The meta-analysis of ST effect was performed on the four studies of children’s RVCs as shown in Figure 2A,B. The standardized correlation coefficients of those studies are shown in Table 2. The association between ST and arteriolar calibers was observed (Figure 2A, r = −0.043, 95%CI −0.078 to −0.009), but no association was found among venular calibers (Figure 2B, r = 0.002, 95%CI −0.044 to 0.047).

The meta-analysis of the three studies was conducted on the effect of vigorous PA on children’s RVCs (Figure 2C,D). Non-significant associations between vigorous PA and CRAE were found (Figure 2C, r = −0.005, 95%CI −0.101 to 0.092; Figure 2D, CRVE, r = −0.003, 95%CI −0.103 to 0.098). The standardized correlation coefficients Sβ of these studies is shown in Table 2.

Separate meta-analyses were conducted on the effects of SB and PA on CRAE and CRVE. We compared the differences of RVCs between the first and fourth quartiles of SB and PA, respectively. We found that the CRAE of the children with more SB was smaller than those with less SB (Figure 3A, CRAE, r = −1.67, 95%CI −3.33 to 0), but there were no significant associations of CRVE with SB (Figure 3B, r = 1.05, 95%CI −3.22 to 5.32). Children with more PA had a larger CRAE (Figure 3C, r = 0.88, 95%CI −3.33 to 0), but no statistical correlation was found among CRVE and PA (Figure 3D, r = −0.07, 95%CI −1.94 to 1.80).

## 4. Discussion

It seemed that the children with more screen time have statistically narrower CRAE. The similar correlation was also found in TV time, which is another measurement for SB besides screen time. The tendency of wider CRVE appeared in the 4th quartile of TV time, but no obvious association of screen time with CRVE was found in the regression models. These findings were similar to those in adults [26,27], which suggested that a reduction in time spent in SB may be important for good retinal vascular profiles. The possible explanation from the physiological aspect may be the lower levels of DNA methylation of p66shc gene, and thus the higher levels of systemic oxidative stress levels in sedentary individuals than healthy active peers [28]. Overall, our study indicated that unhealthy lifestyles might be harmful on the retinal microcirculation among children and adolescents but their health effect seems not to be as significant as those in adults as the effects observed in adults probably requires more than a decade to develop.

The correlation between PA and RVCs appeared more complicated, since the qualification of PA not only involved total time but also activity locations (indoor or outdoor). Children with longer indoor activity tended to have narrower CRAE and CRVE, while the outdoor activity seemed to have the opposite effect, that was wider CRAE and CRVE. The longer total time of PA (including both outdoors and indoors) seemed related to wider CRAE, but no association was found with CRVE. Those statistical findings were found by the comparison between the first and the fourth quartiles of PA, instead of the regression models. Compared to SB, the indoor PA seemed to have similar influence on CRAE. It was suggested that different features between indoor and outdoor environments had potential effects on RVCs, and similar environmental effects on myopia had been reported previously [29]. The findings indicated the potential of outdoor PA as a preventive strategy to improve microvascular and cardiovascular health in children. One possible explanation is the direct hemodynamic impacts on the structure and function of the artery wall resulting from PA [30]. PA can induce the increase of shear stress and thus may increase the NO bioavailability to protect microvascular health. Nonetheless, previous studies in adults found statistical correlations of PA with CRVE rather than CRAE [31]. It may be explained by the inflammation effect on venule calibers [32], since systemic inflammation is an important mechanism of microvascular dysfunction. The PA may benefit the anti-inflammatory mechanics, which were more related to venule rather than arterial profiles [32].

In addition, diet could also contribute to the structural changes of RVCs. The frequent consumption of carbohydrates or soft drinks with high–glycemic load could have harmful effects, leading to the narrower CRAE or wider CRVE (usually considered unhealthy) [9]. One possible explanation of microvascular dysfunction is the acute hyperglycemia from soft drink intakes might impair vasodilatation [33]. On the contrary, the consumption of fish and dairy products could have beneficial effects, particularly among girls, with CRAE widening and CRVE narrowing [7,22]. The pathogenic pathways of diet affection on retinal vessels may be different due to the variety of dietary patterns though. Cumulative adverse effects in dietary trajectories were found in adults, but the associations were not obvious in children, suggesting the window to prevent permanent changes in childhood [10].

Although short sleep durations were found to be related to obesity, its connection with cardiovascular vessels has not been elucidated yet. Only one publication was found to be relevant to sleep duration and AVR for children, and addressed the effect of sleep duration in early childhood on cardiovascular health of later life [26].

Limitations of this analysis should be acknowledged. First, the potential biases in the original studies and methodological issues could affect the results of this meta-analysis. The observed association could have been confounded by other unadjusted factors or selection bias. Second, the number of contributing studies on children and adolescents was small. Therefore, conducting subgroups analysis may not be feasible. Finally, publication bias could be of concern because studies that report statistically significant findings are more likely to be published than those reporting non-significant results, and this could have distorted the findings.

## 5. Conclusions

The results of our meta-analysis demonstrated that the unhealthy life styles might be harmful on the retinal microcirculation among children and adolescents. Although the health effects in our studies were not so strong compared with those observed in adults, healthy lifestyles at early stage of lives might still improve microcirculation and reduce the incidence and associated costs of CVDs in later lives.

## Figures and Tables

**Figure 1 nutrients-15-00150-f001:**
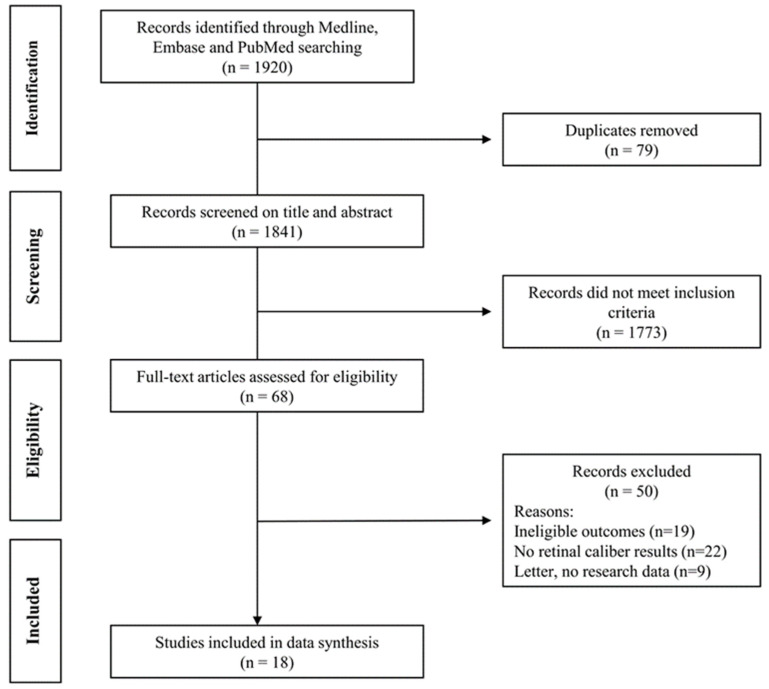
Flow diagram detailing numbers of studies searched, screened and included in the review.

**Figure 2 nutrients-15-00150-f002:**
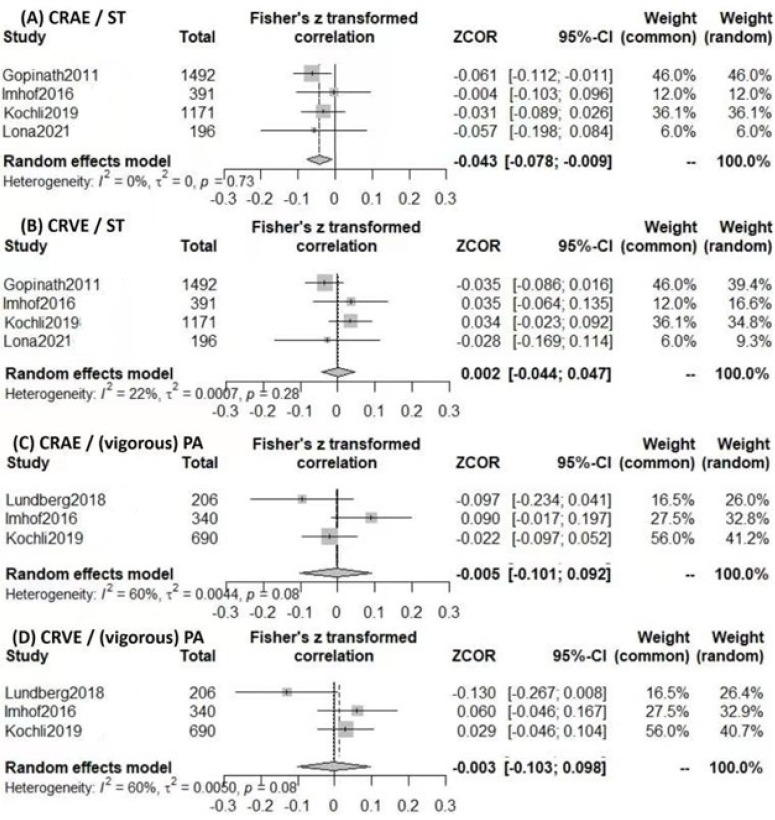
(**A**,**B**) The associations of screen time with retinal vascular calibers for children and adolescents. (**C**,**D**) The associations of vigorous PA with RVC for children and adolescents. CRAE: central retinal arteriolar equivalent; CRVE: central retinal venule equivalent; ST: Screen time; PA: physical activity [3,6,8,11,12].

**Figure 3 nutrients-15-00150-f003:**
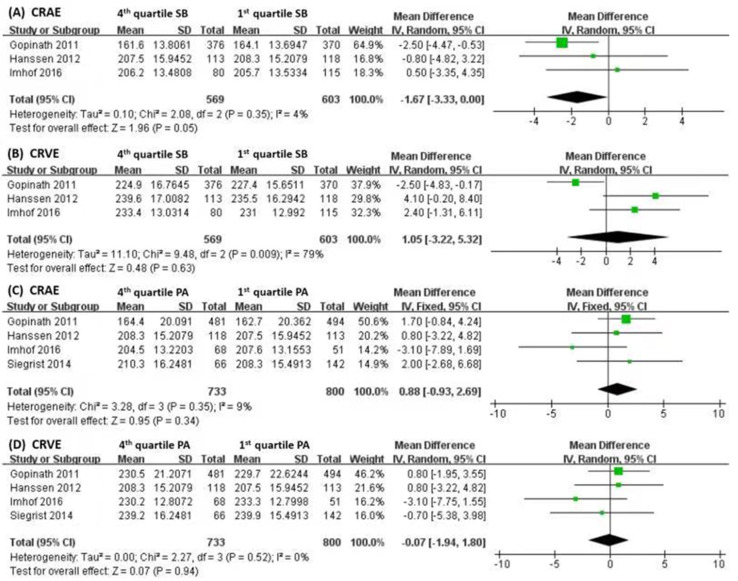
(**A**,**B**) The meta-analysis performed to assess the association between SB (1st and 4th quartiles) and changes in CRAE and CRVE. (**C**,**D**) The meta-analysis performed to assess the associations between PA (1st and 4th quartiles) and changes of CRAE and CRVE. CRAE: central retinal arteriolar equivalent; CRVE: central retinal venule equivalent; SB: sedentary behavior; PA: physical activity [6,11,13,20].

**Table 1 nutrients-15-00150-t001:** Study characteristics of those reporting RVC and lifestyles for children and adolescents.

Study ID	Country	Sample Size (Female%)	Age (Mean, SD/Range)	Exposures	Study Quality (Stars)
Gopinath 2011 [6]	Australia	1492 (49.3)	6.7 (0.4)	PA, SB	7
Hanssen 2012 [13]	Germany	578 (43.1)	11.1 (0.6)	PA, SB	7
Siegrist 2014 [20]	Germany	381 (41.5)	10–11	PA, SB	7
Siegrist 2018 [21]	Germany	434 (42.8)	10–11	PA	8
Lundberg 2018 [12]	Denmark	307 (47.6)	15.4 (0.7)	PA, SB	8
Imhof 2016 [11]	Switzerland	391 (51.2)	7.3 (0.4)	PA, SB	9
Ludyga 2019 [19]	Switzerland	36 (36)	12–15	PA	8
Köchli 2019 [8]	Switzerland	1171 (50.7)	7.2 (0.4)	PA, SB	8
Lona 2021 [3]	Switzerland	262 (54.2)	7.4 (0.3)–11.4 (0.3)	PA, SB	8
Lim 2009 [25]	Singapore	823	12.8 (0.8)	Diet	5
Gopinath 2012 [9]	Australia	1855 (50.6)	12.7 (0.4)	Diet	7
Gopinath 2014 [7]	Australia	888 (49)	girl:12.7 (0.4), boy:12.8 (0.5)	Diet	7
Gopinath 2017 [22]	Australia	1920 (50.7)/ 1199 (55.3)	12/17	Diet	7
Kerr 2018 [23]	Australia	188 (52)	15.1 (0.5)	Diet	6
Davis 2019 [24]	Australia	1771 (49)	11–12	Diet	7
Saraf 2022 [16]	Australia	1838 (49.1)	11.5	Diet	7
Kerr 2021 [10]	Australia	1861 (49)	2–11	Diet	7
Derks 2019 [26]	Australia	336 (50.6)	14	Sleep	6

**Table 2 nutrients-15-00150-t002:** Studies reported associations between PA/SB/Sleep and RVC.

Variable	Study ID	Statistical Analysis	Minimum Level of Adjustment		Associations	
				Arteriole CRAE	Venule CRVE	AVR
PA						
	Siegrist 2014 [20]	LR	Age and gender	*Sβ* = 0.098 (*p* = 0.492)	*Sβ* = −0.11 (*p* = 0.436)	*Sβ* = 0.19 (*p* = 0.171)
	Siegrist 2018 [21]	LR	Study group, age and gender	*β* = 0.019 (PA_school_, *p* = 0.284)	*β* = −0.001 (*p* = 0.967)	None
	Lona 2021 [3]	LR	Age, sex, height	*Sβ* = −0.055 (*p* = 0.934)	*Sβ* = −0.06 (*p* = 0.271)	*Sβ* = −1.5 (*p* = 0.189)
	Ludyga 2019 [19]	ANCOVA		A structured exercise program lead to a widening CRAE (*p* = 0.036)	No effect was observed on CRVE.	None
Vigorous PA						
	Lundberg 2018 [12]	LR	Age, gender and axial length	*Sβ* = −0.040 (*p* = 0.490)	*Sβ* = −0.16 (*p* < 0.010)	*Sβ* = 0.100 (*p* = 0.110)
	Imhof 2016 [11]	LR	Age and sex	*Sβ* = 0.090 (*p* = 0.200)	*Sβ* = 0.060 (*p* = 0.200)	*Sβ* < 0.820 (*p* = 0.800)
	Köchli 2019 [8]	LR	Age and sex	*Sβ* = −0.045 (*p* = 0.224)	*Sβ* = 0.022 (*p* = 0.560)	*Sβ* = −0.100 (*p* = 0.008)
SB(/ST/TV)						
	Hanssen 2012 [13]	LR	Age and gender	*Sβ* = −0.022 (*p* = 0.589)	*Sβ* = 0.073 (*p* = 0.045)	*Sβ* < 0.260 (*p* = 0.010)
	Lundberg 2018 [12]	LR	Age, gender, and axial length	*Sβ* = 0.037 (*p* = 0.54)	*Sβ* = 0.160 (*p* < 0.01)	*Sβ* = −0.087 (*p* = 0.050)
	Gopinath 2011 [6]	LR	Age, sex, ethnicity, iris color, axial length, BMI, birth weight and MABP	*Sβ* = −0.061(ST, *p* = 0.02)	*Sβ* = 0.350 (*p* = 0.080)	None
				*Sβ* = −0.066 (TV, *p* = 0.006)	*Sβ* = −0.020 (*p* = 0.280)	None
	Imhof 2016 [11]	LR	Age and sex	*Sβ* = −0.035 (ST, *p* = 0.500)	*Sβ* = 0.035 (*p* = 0.700)	*Sβ* <0.960 (*p* = 0.200)
	Köchli 2019 [8]	LR	Age and sex	*Sβ* = −0.058 (ST, *p* = 0.089)	*Sβ* = 0.026 (*p* = 0.435)	*Sβ* = −0.120 (*p* = 0.002)
	Lona 2021 [3]	LR	Age, sex, height	*Sβ* = −0.038 (ST, *p* = 0.075)	*Sβ* = −0.041 (*p* = 0.421)	*Sβ* = −1.100 (*p* = 0.359)
Diet						
	Lim 2009 [25]	LR	Age, gender, MABP and BMI	*β* = 0.060 (fiber, *p* = 0.350), *β* = 0.005 (sugar, *p* = 0.710)	*β* = −0.010 (fiber, *p* = 0.950)*β* = −0.010 (sugar, *p* = 0.640)	None
	Gopinath 2012 [9]	ANCOVA		Great consumption of soft drinks and cordials narrowed retinal arterioles (*p* = 0.030).	High carbohydrate consumption widened venules among boys (*p* = 0.020).	None
	Gopinath 2014 [7]	ANCOVA		Yogurt intake widened retinal arterioles of adolescents (*p* = 0.050)	Yogurt intake narrowed venules of adolescents (*p* = 0.040).	None
	Gopinath 2017 [22]	ANOVA		LCn-3PUFA intake widened the retinal arteriolar caliber among girls but not boys (*p* = 0.001).	No associations were observed with retinal venules.	None
	Davis 2019 [24]	LR	Age, sex and SEP	*β* = 0.340 (inflammatory diet score, *p* = 0.330)	*β* = 0.210 (*p* = 0.670)	None
	Saraf 2022 [16]	LR	Age, sex and SEP	*Sβ* = 0.130 (takeaway food, *p* = 0.050)	*Sβ* = 0.030 (*p* = 0.650)	None
				*Sβ* = 0.090 (SSB, *p* = 0.190)	*Sβ* = 0.040 (*p* = 0.510)	None
	Kerr 2021 [10]	LR	Age, sex and SEP	*Sβ* = −0.050 (DQ moderately healthy, *p* = 0.540)	*Sβ* = 0.020 (*p* = 0.790)	None
				*Sβ* = 0.040 (DQ less healthy, *p* = 0.550)	*Sβ* = −0.030 (*p* = 0.680)	None
				*Sβ* = 0.130 (DQ never healthy, *p* = 0.320)	*Sβ* = −0.090 (*p* = 0.420)	None
	Kerr 2018 [23]			Decade-long dietary trajectories did not appear to influence microvascular structure by mid-adolescence		
Sleep						
	Derks 2019 [26]			Infant sleep duration was not associated with artery structure at 14 years of age.		

CRAE: central retinal arteriolar equivalent; CRVE: central retinal venule equivalent; AVR: arteriole to venule ratio; LR: linear regression; SEP: socioeconomic position; ANOVA: analysis of variance; ANCOVA: analysis of covariance; SB: sedentary behavior; ST: Screen time; TV: television (time); MABP: mean arterial blood pressure; BMI: Body mass index; SMD: standardized mean difference; SSB: sugar-sweetened beverage; DQ: diet quality.

## Data Availability

Not applicable.

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
