# Peer review of "Unhealthy Lifestyles and Retinal Vessel Calibers among Children and Adolescents: A Systematic Review and Meta-Analysis"

_nutrients, 2022, doi:10.3390/nu15010150_

Round 1

Reviewer 1 Report

Title:  Unhealthy lifestyles and retinal vessel calibers among children and adolescents: a systematic review and meta-analysis.

Authors: Li, Dan-Lin, Zhou, Mia, Chen, Dan-Dan, Liu, Meng-Jiao, Pan, Chen-Wei, and * (It appears that an author is missing on the author list as there is no name after the and….either the and * needs to be removed or a final author needs to be listed.)

Summary: Studies on lifestyles and retinal vessels on children and adolescents are limited compared to studies on adults. Retinal vessel caliber (RVC) is a biomarker of cardiovascular diseases which can be measured by fundus photography. This paper is a review of investigations that evaluate RVCs and different lifestyles early in life in studies of children and adolescents suggesting that unhealthy lifestyles could have harmful effects on retinal microcirculation and that this could be used as a biomarker in children and adolescents; however, the main statistical analysis showed that this was not statistically significant; therefore, would likely not be useful as a biomarker for cardiovascular disease in children. The study included a small number of papers which is a limitation of this study. There were only a few studies of diet and those came from different countries, where diets are likely different and exercise is probably different too; therefore, not directly comparable. The effects seen in adults likely requires more than a decade to develop and this might be a point to make in the discussion as to why this is a biomarker in adults but why you don’t see the same thing in children and adolescents.  There may be a better place to publish this such as  a journal that focuses on bioinformatics.

Revisions needed:

Page 2, Line 46 – Change the word diagnose to diagnostic.

Page 2, Lline 51 – Delete the word an before increasing.

Page 2, Line 56 – Change the word consumptions to consumption.

Page 2, Line 58 – Change the word these before existing to the.

Page 2, Line 59 – Insert the word the before early stage.

Page 2, Line 61 – Change the word dietary to diet.

Page 3, Line 128 – Change the word dietary to diet.

Page 3, Line 133 – Insert the word the before PA.

Page 5, Line 142 – Change the word dietary to diet.

Page 9, Line 162 – Delete the word were before addressed.

Page 9, Line 163 – Delete the word on after addressed.

Page 9. Line 166 – Insert the word a before long.

Page 9. Line 167 – Change the word teenage to teenager.

Page 9, Line 177 – Change had been to is.

Page 9, Line 205 – Change the word potentials to potential.

Page 9. Line 205 – Insert the word a before preventive.

Page 9, Line 205 – Change strategies to strategy.

Page 9, Line 208 – Change resulted to resulting.

Page 9, Line 211 – Remove the word the before systemic.

Page 10, Line 228 – Remove the word on after addressed.

Page 10, Line 229 – Remove the before cardiovascular.

Page 10, Line 231 – Change issue to issues.

Page 10, Line 233 – Remove the before contributing.

Page 10, Line 241 – Change reduced to reduce.

Reviewer 2 Report

This is a very interesting work. My only concern is about the conclusion vs the analysis provided. I would be way more nuanced about the fact that this analysis proved a solid link between lifestyle and retinal vasculature. The analysis is bringing some - slightly- significance for arterial vessels but not for veins.  I would be prudent to conclude that lifestyle influences the retinal vasculature. There are some indications but it is not so obvious.  Just like if the authors would want to make their point, - preconceptual point- and justify it by an analysis that is not so strong.     I strongly recommend to tone down the article in this regard

Author Response

We thank the reviewer for highlighting this issue. We now tone down conclusion in abstract and discussion. We acknowledge that the effect size found in our study was not so strong as compared with that in adult study. We hope the reviewer could be satisfied with the revised version.